# Muscular Strength Imbalances Are not Associated with Skin Temperature Asymmetries in Soccer Players

**DOI:** 10.3390/life10070102

**Published:** 2020-07-02

**Authors:** Rodrigo Mendonça Teixeira, Rodolfo A. Dellagrana, Jose I. Priego-Quesada, João Claudio B.P. Machado, Juliano Fernandes da Silva, Tallyne Mayara Pacheco dos Reis, Mateus Rossato

**Affiliations:** 1Human Performance Laboratory, Federal University of Amazonas, Manaus 69067-005, Brazil; rodrigomendon@hotmail.com (R.M.T.); joaoclaudiomachado@gmail.com (J.C.B.P.M.); mateusrossato@ufam.edu.br (M.R.); 2Graduate Program in Movement Sciences, Federal University of Mato Grosso do Sul, Campo Grande 79070-900, Brazil; radellagrana@gmail.com; 3Research Group in Sports Biomechanics (GIBD), Department of Physical Education and Sports, University of Valencia, 46010 Valencia, Spain; 4Biophysics and Medical Physics Group, Department of Physiology, University of Valencia, 46010 Valencia, Spain; 5Research Center for Development of Football and Futsal, Federal University of Santa Catarina, Florianópolis 88040-900, Brazil; jufesi23@yahoo.com.br; 6Graduate Program in Health Sciences, Federal University of Amazonas, Manaus 69067-005, Brazil; tpachecofisio@gmail.com

**Keywords:** injury, prevention, thermal image, strength imbalance, football, thermography

## Abstract

Although strength imbalances using isokinetic dynamometer have been examined for injury risk screening in soccer players, it is very expensive and time-consuming, making the evaluation of new methods appealing. The aim of the study was to analyze the agreement between muscular strength imbalances and skin temperature bilateral asymmetries as well as skin temperature differences in the hamstrings and quadriceps. The skin temperature of the anterior and posterior thigh of 59 healthy male soccer athletes was assessed at baseline using infrared thermography for the identification of hamstrings-quadriceps skin temperature differences and thermal asymmetries (>0.5 °C). Subsequently, concentric and eccentric peak torque of the quadriceps and hamstrings were considered in the determination of the ratios, as well as muscular asymmetries (>15%). When considering the torque parameters, 37.3% (*n* = 22) of the players would be classified as high risk for injuries. The percentage of those presenting skin temperature imbalances superior to 0.5 °C was 52.5% (*n* = 31). The skin temperature assessment showed sensitivity (22%) and specificity (32.2%) to identify torque asymmetries, demonstrating the inability to identify false negatives (15.3%) and false positives (30.5%) from all soccer athletes. In conclusion, skin temperature differences between hamstrings and quadriceps could be more related to thermoregulatory factors than strength imbalances.

## 1. Introduction

Soccer presents a high rate of skeletal-muscle injuries in comparison with other sports, which can result in elevated medical leave rates during matches and training [1,2]. Among them, hamstring injuries caused by indirect mechanism (i.e., no contact injury) [3,4] have been a problem in male soccer players [3,5,6]. In general, these injuries commonly occur at the time of eccentric action to the hamstrings (deceleration of the lower limb in the late swing phase) and quick changes from eccentric to concentric action, mainly when the hamstrings become active extensors of the hip joint [5,7,8]. Therefore, strength imbalances have been assessed for injury risk screening in soccer players [3,4,5,6,9].

For these athletes, the diagnostics of side-to-side strength asymmetries and muscle imbalance between the flexor and extensor were assessed by an isokinetic dynamometer [4,10,11], which was proven valid and reliable [12]. Croisier et al. [5] evaluated 462 soccer players during pre-season, where the authors observed that untreated athletes with muscular deficits (asymmetry and muscle imbalances) presented more than fourfold the risk of hamstring injuries compared to players without any preseason muscular deficits. Similarly, Lehance et al. [6] noticed that 64% of the previously injured soccer players continued to exhibit muscle imbalances, indicating a high risk of another injury. These findings reinforce that the preseason isokinetic evaluation in soccer players is important to identify muscle deficits. On the other hand, isokinetic evaluations are very expensive and time-consuming, diminishing the sports training time [5,6]. Thus, innovative strategies that are reliable and expeditious in the identification of muscular deficits can be useful for team sports.

Recently, several studies have used infrared thermography (IRT) in the medical diagnosis for different pathologies [13,14,15]. IRT is a non-invasive technique that is low-cost (i.e., compared to isokinetic dynamometer), painless, contactless, non-ionizing radiation and innocuous; besides, it allows the evaluation of skin temperature (T_skin_) in real-time [13,16,17,18]. Studies have suggested that bilateral differences greater than 0.5 or 0.7 °C have been associated with physiological abnormalities [19,20,21]. In the sports scenario, IRT assessed at rest has been frequently used to pinpoint the injury location through skin temperature asymmetries [19,22,23], which could be mainly associated with inflammation or alteration of blood perfusion processes [17,24]. 

The aims of this study were: (a) to identify bilateral strength asymmetries and muscular imbalances of the hamstring and quadricep muscles of soccer players; (b) to identify skin temperature bilateral asymmetries of the aforementioned muscles and hamstrings and quadriceps skin temperature differences; and (c) to assess the agreement between strength imbalances and skin temperature parameters. It was hypothesized that athletes classified as high risk for hamstring injuries by isokinetic parameters could present high bilateral and hamstrings-quadriceps asymmetries in skin temperature.

## 2. Materials and Methods 

### 2.1. Participants

Fifty-nine healthy male soccer athletes (19.7 ± 3.3 years, 68.8 ± 9.0 kg and 10.3 ± 4.4% body fat) participated in the present study. Lower-limb preference was determined by asking them, “If you were to shoot a ball at a target, which leg would you use?” [25]. Forty-five players preferred the right lower limb. All participants gave their written informed consent before participation, and the study was approved by the ethics committee of the university in agreement with the Declaration of Helsinki (CAAE: 56226716.7.0000.5020) on 25 July 2016. 

To reduce skin temperature variability among the participants and to obtain reliable measurements, they were instructed to [26]: (a) avoid high-intensity exercise the day before the test; (b) not drink alcohol, coffee, stimulant drinks or smoke 12 h before the test; (c) not sunbathe or be exposed to UV rays 24 h before the test; (d) avoid body lotions and cream; and (e) eat at least 2 h before the test and refrain from having a heavy meal.

### 2.2. Thermography Data Collection and Analysis

The protocol involved the acquisition of infrared thermography images, after participants arrived at the laboratory, under controlled conditions of temperature (23.9 ± 1.4 °C) and relative humidity (49.8 ± 2.25%). The T_skin_ was measured by a thermography camera with an infrared resolution of 320 × 240 pixels and a thermal sensitivity of 0.045 °C (T420, FLIR, Wilsonville, OR, USA). The Thermal Imaging in Sports and Exercise Medicine (TISEM) checklist was used to ensure that all the important aspects related to thermographic measurements were verified [14]. Before the acquisition of the thermal images, participants remained standing at rest wearing only underwear for 10 min of their thermal adaption to the room [27]. Moreover, the thermal camera was turned on 10 min before measurements to ensure its stabilization [28]. Afterward, thermal images of the thighs were recorded. The thermal images were taken perpendicular to the thighs while participants were standing wearing only underwear, at a distance of 1 m from the camera. To ensure the quality and reproducibility of the thermal image, it was taken with the lights off, with only the thermography technician and the participant in the measurement room. No electronic equipment was located within a 5 m of the measurement space, and an anti-reflective panel was placed behind the participant to avoid the effects of radiation reflected by the wall [13,14,17]. 

Air temperature, relative humidity and reflected temperature were measured and included in the camera settings. Two regions of interest (ROIs) were defined in the thermal images of the anterior (quadriceps) and posterior thighs (hamstrings), in both lower limbs, preferred (PL) and non-preferred lower limb (NPL) (Figure 1). Mean skin temperature was obtained using thermography software (Thermacam Researcher Pro 2.10, FLIR, Wilsonville, USA). All analyses were performed with a skin emissivity of 0.98 [29]. Regarding skin temperature differences, at least one of the following parameters was used to identify imbalances: bilateral (PL-NPL) and hamstrings–quadriceps differences greater than 0.5 °C. In a previous study, 0.4 °C was determined as the maximum temperature symmetry difference in healthy subjects, thus 0.5 °C was considered the cut-off for temperature imbalance [20].

### 2.3. Isokinetic Data Collection and Analysis

Isokinetic collection and analysis were performed after the acquisition of the thermal images according to the methods and recommendations described by Croiser et al. [5]. An isokinetic dynamometer, Biodex System 4.0 (Biodex Medical Systems, Shirley, NY, USA) was used to test the concentric and eccentric torque of the knee extensor and flexor muscles. Prior to the test, the device was calibrated following the manufacturer’s recommendations. A warm-up was performed in a cycle-ergometer with a load of 75–100 W for 5 min. Subsequently, the evaluator explained the testing procedures in detail and placed the volunteer on the equipment seat, at an angle allowing the hip joint to be at 105° flexion, with the body stabilized by straps around the thigh, waist and chest to avoid compensation. The range of knee motion was fixed at 100° flexion from the active maximum extension (Figure 2). 

The rotational axis of the dynamometer arm was aligned with the lateral epicondyle of the femur of the right and left lower limb. The site of force application was positioned approximately 2 cm from the medial malleolus. Belts were fixed to the trunk, pelvis and thigh to prevent compensatory movements. The concentric torque of the quadriceps and hamstrings were evaluated in 3 repetitions at 60°/s and in 5 repetitions at 240°/s. The eccentric torque of the hamstrings was evaluated in 3 repetitions at 30°/s and in 4 repetitions at 120°/s. The recovery interval time between sets was 1 min. The same procedure was performed for the left lower limb. The volunteers were instructed to perform maximal strength. During the test, the participants were strongly encouraged, verbally, to exert “harder” and “stronger” efforts. 

The results analyses were expressed in absolute (N.m) concentric and eccentric Peak Torque (PT) of knee extensors and flexors, as well as the bilateral comparison (preferred and non-preferred limbs), led to the determination of asymmetries [30]. The concentric H/Q peak torque ratio of flexors and extensors was established (at 60°/s or 240°/s) and the mixed Hecc/Qconc ratio was associated with the eccentric performance of the hamstrings and the concentric action of the quadriceps muscles (hamstrings at 30°/s versus quadriceps at 240°/s). Regarding the strength imbalance profile, this study followed the procedures described by Crosier et al. [5]: bilateral differences above 15% in concentric and/or eccentric in the hamstrings; concentric ratio (in at least 1 leg) of less than 0.45; and a mixed ratio of less than 0.89 on Biodex. Thus, at least 2 of the following parameters were used to identify the injury risk: concentric (at 60°/s or 240°/s) and eccentric (at 30°/s or 120°/s) bilateral asymmetries (>15%); conventional Hconc/Qconc (at 60°/s or 240°/s); and mixed Hecc/Qconc ratio.

### 2.4. Statistical Analyses

Data are presented as means, standard deviation and frequencies. The Kolmogorov–Smirnov test was used to check the normality of the data distribution. Torque and temperature ratios for preferred and non-preferred lower limbs were compared implementing independent t-tests. Two-way analysis of variance (group factor (high and low risk of strength imbalances) and limb factor (preferred and non-preferred)) was applied to compare muscular group (quadriceps and hamstrings) temperature. The Bonferroni post hoc was used for all analyses. All tests were performed using SPSS Statistics for Windows, version 21.0 (SPSS Inc., Chicago, IL, USA). The significance level was set at 0.05 for all comparisons. 

## 3. Results

NPL hamstrings presented lower values (Table 1) for PT in three of the four velocities evaluated (30°/s and 120°/s eccentric and 240°/s concentric). In addition, the NPL exhibited statistically lower values for the conventional ratio (Conc/Conc 240°/s) and mixed ratio (Ecc 30°/s/Conc 240°/s). 

No significant differences were found between PL and NPL for the skin temperature of the quadricep and hamstring ROIs between the high and low risk groups of muscular imbalances (Table 2).

The highest rates of players with bilateral differences in PT were observed at velocities of 30°/s eccentric (30.5%), 60°/s concentric (23.7%) and 120°/s eccentric (22.0%) (Table 3). Regarding t_skin_ differences higher than 0.5 °C, higher rates of players were observed when considering the difference between hamstrings and quadriceps (28.8% for PL and 44.1% for NPL) than for the difference between the preferred and non-preferred limb (3.9% for quadriceps and 0% for hamstrings). In terms of torque parameters, 37.3% (*n* = 22) of those evaluated would be classified as having high risk of injuries. On the other hand, 52.5% (*n* = 31) of those assessed presented thermal imbalances above 0.5 °C.

The IRT showed sensitivity (capacity to identify strength imbalance) and specificity (strength balance) of 22% and 32.2%, respectively, demonstrating the inability to identify false negatives (t_skin_ balance and strength imbalance) and false positives (t_skin_ imbalance and strength balance) of 15.3% and 30.5%, respectively (Table 4).

## 4. Discussion

This study aimed to identify asymmetries between limbs and muscular imbalances of soccer players and to associate t_skin_ asymmetries through IRT. We hypothesized that athletes classified in the hamstring injury risk group by isokinetic evaluation would also present asymmetries in skin temperature parameters. However, the results indicate that the t_skin_ asymmetry was not related to the strength imbalances indicated in the isokinetic test with cut-off points and normative data [5]. To the best of the authors’ knowledge, this is the first study to associate dynamometric parameters that are indicative of a high risk of hamstring injuries in soccer players with t_skin_ through IRT. 

Considering torque asymmetries, significant differences were observed between preferred and non-preferred limbs for hamstring concentric contractions at 240°/s and eccentric contractions at 30°/s. Similarly, bilateral asymmetries for soccer and futsal players were observed by other studies as well [3,4,5,6,31,32]. Ruas et al. [31] showed that the preferred limb of soccer players presented higher eccentric hamstring strength than the non-preferred limb. In addition, Nunes et al. [32] demonstrated that futsal players had greater concentric (at 240°/s) and eccentric (at 30°/s and 120°/s) hamstring strength for the preferred limb in comparison to the non-preferred limb. Despite these findings in the current study, the group average asymmetry did not exceed 15% (Table 1); however, across all torque categories (Conc at 60°/s and 240°/s; Ecc at 30°/s and 120°/s), only 10.2–30.5% of players exhibited asymmetries above 15% (Table 3). As a possible explanation, the non-preferred limb plays an important role in supportive strength to coordinate dominant knee actions [31].

In the present study, according to Croisier et al. [5], strength imbalances assessed by conventional and functional ratios presented low incidence (Table 3). In soccer players, the hamstrings muscle group is the most affected by injuries [33]. The hamstring muscles have a significant function in decelerating the extension of the lower limb in the thigh during ball striking, which can harm or damage the muscle–tendon unit [34,35]. In addition, hamstring muscles can be vulnerable to injury usually during quick changes from eccentric to concentric action, especially when the hamstrings become hip joint extensors [5,36]. Therefore, the factors responsible for the high incidence of hamstring injuries are muscle weakness, strength imbalance and previous injuries [5,6]. In an attempt to reduce the number of injuries in this muscle, recent studies have shown positive results with the inclusion of eccentric training sessions in the players’ routine [37,38].

Regarding t_skin_ evaluated by IRT, no significant differences were found between the quadricep and hamstring ROIs, regardless of the injury risk or limb preference (Table 2). Bilateral t_skin_ asymmetries in the quadricep and hamstring ROIs presented low incidence. Similarly, Bouzas Marins et al. [39] also reported the absence of bilateral t_skin_ asymmetries for soccer players. These results are in agreement with other studies that observed the differences in the application of forces between lower limbs did not result in t_skin_ asymmetries [40,41]. Hence, bilateral t_skin_ asymmetries could be more related to the effect of an injury (e.g., inflammatory process or alteration of blood perfusion) [17,22] than with muscular strength imbalance. 

On the other hand, a high incidence of thermal imbalances (>0.5 °C) was observed for hamstrings–quadriceps t_skin_ of PL (28%) and NPL (44%) (Table 3). In this context, some investigations showed contradictory results. While Chudecka and Lubkowska [42] reported superior t_skin_ in the quadriceps compared to hamstring ROIs, these findings were not confirmed by Bouzas-Marins et al. [39]. The causes for the presence of a higher local t_skin_ are generally linked to the occurrence of hyperemia due to the inflammation process caused by the increase in blood flow to the injured region [43]; nonetheless, in the current study, no athlete reported injury at the time of evaluation. This data also indicate that the assessment of t_skin_ by IRT when resting does not seem to be able to screen strength imbalances (positive-false = 30.5% and negative-false = 15.3%), resulting in hamstring injury risk. Therefore, it can be assumed that thermal differences between quadriceps and hamstrings could be more of a result of differences in tissue proportion (e.g., body fat), blood perfusion and capacity of heat dissipation than muscular strength imbalance. 

Some limitations could be associated to the study. A higher sample with female players and a larger age range would have allowed the analysis of gender and age effects. Moreover, the measurement of other physiological parameters, such as muscle damage, neuromuscular activation or muscle oxygenation, would assist in interpreting the results as well. However, this study pioneered the investigation of possible relationships among the parameters commonly used for the identification of injury risk factors for the thigh muscles (dynamometric parameters) of soccer players and t_skin_ by IRT of hamstrings and quadriceps. Considering previous reports about the importance of a daily IRT asymmetry assessment in injury prevention [17,19,21,22,23], long-term follow-up studies of athletes using IRT evaluation are encouraged. Finally, based on a recent study [44], further investigations assessing the skin temperature recovery after a cold stress test could add valuable information to the vascularization status of the region analyzed by infrared thermography.

## 5. Conclusions

The evaluation of skin temperature asymmetries by IRT at rest was unable to identify strength imbalances, and, consequently, hamstring injury risk in soccer players. Thermal differences between hamstrings and quadriceps could be more related to thermoregulatory factors than with strength imbalances. 

## Figures and Tables

**Figure 1 life-10-00102-f001:**
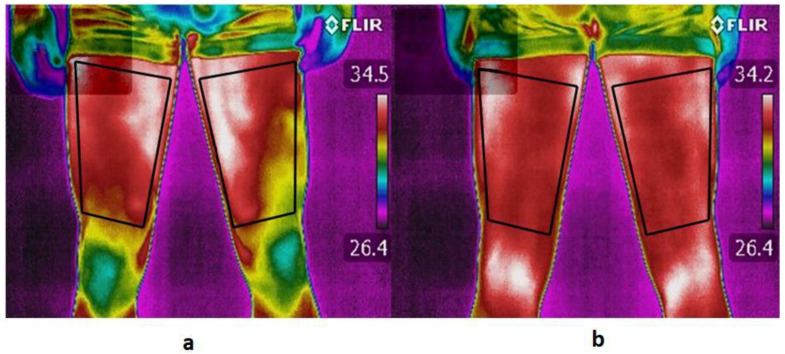
Determination of the ROIs of the anterior thigh (**a**) and posterior thigh (**b**) in both lower limbs. ROI was determined to cover the entire thigh, being the upper definition of the crotch, and the lower definition of the knee or popliteus.

**Figure 2 life-10-00102-f002:**
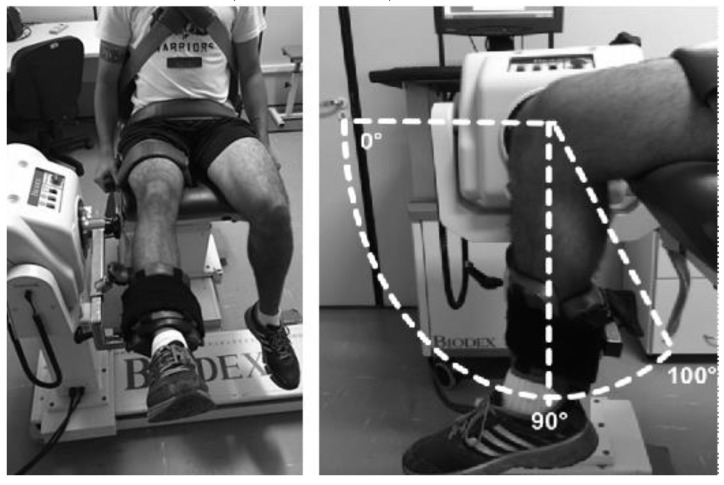
Participant positioning for the isokinetic dynamometry test.

**Table 1 life-10-00102-t001:** Mean ± standard deviation PT and the hamstrings-to-quadriceps ratio (H/Q ratio) in preferred (PL) and non-preferred (NPL) limbs and asymmetry index (AI).

Isokinetic Variables	PL (N^.^m)	NPL (N^.^m)	AI (%)	*p*
**Quadriceps**				
60°/s Conc.	206 ± 36	196 ± 36	3.99 ± 13.7	0.15
240°/s Conc.	127 ± 21	128 ± 26	−1.04 ± 18.0	0.93
**Hamstrings**				
30°/s Ecc.	172 ± 31	160 ± 31	6.62 ± 14.3	0.03 *
60°/s Con.	115 ± 23	114 ± 28	0.81 ± 26.1	0.85
120°/s Ecc.	173 ± 33	162 ± 27	4.66 ± 15.5	0.04 *
240°/s Con.	84 ± 15	78 ± 14	6.77 ± 11.7	0.02 *
**H/Q Ratio**				
Conc/Conc 60°/s	0.56 ± 0.09	0.59 ± 0.16	-	0.22
Conc/Conc 240°/s	0.66 ± 0.08	0.62 ± 0.09	-	0.006 *
Ecc30°/s/Conc240°/s	1.37 ± 0.24	1.27 ± 0.24	-	0.03 *

* Significant differences between PL and NPL.

**Table 2 life-10-00102-t002:** Skin temperature in the quadriceps and hamstrings for preferred (PL) and non-preferred limb (NPL) for the groups of high-risk torque imbalances (*n* = 22) and low-risk torque imbalances (*n* = 37).

Variables	Imbalance Risk	PL (°C)	NPL (°C)	F ^1^	*p*
Quadriceps	High	32.51 ± 0.81	32.49 ± 0.76	0.002	0.964
	Low	32.38 ± 0.82	32.37 ± 0.79		
Hamstrings	High	32.32 ± 0.77	32.35 ± 0.75	0.002	0.962
	Low	32.29 ± 0.68	32.30 ± 0.69		
∆Temp (H-Q)	High	−0.19 ± 0.51	−0.13 ± 0.49	0.022	0.882
	Low	−0.10 ± 0.46	−0.07 ± 0.43		

^1^ Anova interaction.

**Table 3 life-10-00102-t003:** Rate of soccer players with torque and temperature imbalances criteria.

Variables	Rate of Players (%)
**Bilateral Difference**	
Conc 60°/s	14/59 (23.7)
Conc 240°/s	6/59 (10.2)
Ecc 30°/s	18/59 (30.5)
Ecc 120°/s	13/59 (22.0)
∆ Temp. (Quadriceps PL–Quadriceps NPL)	2/59 (3.9)
∆ Temp. (Hamstring PL–Hamstring NPL)	0/59 (0.0)
**Preferred Limb**	
Conc 60°/s/Conc 60°/s	5/59 (8.5)
Conc 240°/s/Conc 240°/s	0/59 (0.0)
Mixed Ecc 30°/s/Conc2 40°/s	1/59 (1.7)
∆Temp. (Hamstrings–Quadriceps)	17/59 (28.8)
**Non-preferred Limb**	
Conc 60°/s/Conc 60°/s	6/59 (10.2)
Conc 240°/s/Conc 240°/s	2/59 (3.4)
Mixed Ecc3 0°/s/Conc 240°/s	4/59 (6.8)
∆ Temp. (Hamstrings–Quadriceps)	26/59 (44.1)
**Injury Criteria**	
Deficiency at least 2 parameters	22/59 (37.3)
Total difference of temperature	31/59 (52.5)

Conc, concentric; Ecc, eccentric; H/Q, hamstring/quadriceps ratio; P, preferred limb; NP, non-preferred limb.

**Table 4 life-10-00102-t004:** Prevalence of torque and skin temperature asymmetries, differences, sensitivity, specificity, false-positive and false-negative rates of infrared thermography to identify torque asymmetries observed by the isokinetic method.

Frequency Analysis	Overall
Strength imbalance (%)	37.3
Skin temperature imbalance (%)	52.5
Difference (%)	15.2
Sensitivity (%)	22.0
Specificity (%)	32.2
Negative-false (%)	15.3
Positive-false (%)	30.5

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
