# Peer review of "Muscular Strength Imbalances Are not Associated with Skin Temperature Asymmetries in Soccer Players"

_life, 2020, doi:10.3390/life10070102_

Round 1
Reviewer 1 Report
Abstract:
The aim of the study is not clear and should be rewarded to cover with the abstract in the introduction part.
Introduction:
Lines 63-64 I do not understand why authors hypothesized that that IRT associated with isokinetic test could be a useful tool to identify risk of hamstrings injuries pre-training.” Before lines 52-54 authors wrote that isokinetic evaluation is very expensive and time consuming thus, valid and fast innovative strategies to identify muscular deficits can be useful for team sports. Therefore authors should hypothesis agreement isokinetic evaluation and skin temperature parameters. I would delete that sentence.
Lines 65-66 aim of the study should be rewarded: The aims of this study were: a) to identify strength side-to-side asymmetries and muscular imbalances of soccer players; What are: side-to-side asymmetries
- b) to identify skin temperature bilateral asymmetries and hamstrings quadriceps differences; I suggest to write: identify skin temperature bilateral asymmetries of hamstring and quadriceps muscles
Lines 68-70: The hypothesis is not clear to me. What is: “side-to-side and agonist- antagonist asymmetries in skin temperature”. Also I do not see a reason to put references in the hypothesis. That reference can be included above the aim of the study.
Materials and methods.
It is not included how participants of the study were prepared for thermographic examination. What the did before the 10 minutes of acclimatization ? where they at rest ? 10 min is not enough to get rid of increased blood circulation in response to any physical exercises? what they were wearing before acclimatization time ? How the examination room was prepared for thermographic examination?
Line 92 I do not understand what authors meant by writing: “both lower limbs, preferred (PL) and non-preferred limb (NP)” that should be explained that this refers to players’ limb preference
Lines 94-95 information about the emissivity should be together with information about the distance between camera and participant.
Lines: 95-96 why skin temperature imbalance? Shouldn’t be skin temperature differences?
When Isokinetic data was collected?
Line 188: What kind of asymmetries between limbs?
Reviewer 2 Report
General comments
This interesting study relates strength imbalances in the thighs of soccer players to skin temperature differences, finding no relationship. The study is well designed with regard to the isokinetic measurements and thermography laboratory protocols. However, the study may be underpowered, and produces a Kappa for agreement between isokinetic injury risk factors and temperature differences which is not significant. Therefore, the authors should consider reformulating the paper as a descriptive report, rather than applying a statistical analysis which is inappropriate.
The English through most of the manuscript requires some improvements, and I would recommend that a native English speaker works with the authors to revise the paper.
Abstract
Line 22-23: It is not clear how thermography was performed in “the anterior and posterior regions…” “…of the anterior thigh”. Please revise this statement to improve the clarity.
Line 31: Do the values given here represent positive and negative predictive values, or simply the proportion of false positives/negatives from all cases? It would be helpful to make this clearer.
Introduction
Line 67: Revise “hamstrings-quadriceps differences” to “hamstrings-quadriceps temperature differences” to improve clarity.
Line 69-70 Not all readers will understand the terminology of “agonist v antagonist” muscle function, and this should be explained.
Materials and methods
Line 73: It needs to be explained why some of your “U-20” players were over the age of 20 years.
Line 83: Give more details about how long the camera was operated for before use. Was camera performance investigated with a quality assurance programme? The FLIR T420 is an uncooled thermal imager, and does not feature a condenser.
Line 86: Why was it necessary to switch off the laboratory lights? The thermal imager is sensitive to a 7-14 microns waveband, so visible light would not affect the signal.
Line 87: Explain why no electronic equipment was within 5m? Were there problems from interference with any of the measurement equipment?
Line 97: Explain why chose you chose 0.5C as the cut-off value for temperature imbalance. Is there a compelling reference that you relied upon?
Line 103-107: I would suggest moving this section to the top of section 2.2, as these are requirements that had to be adhered to prior to any measurements.
Results
In Table 4, the Kappa statistic not significant: this suggests that the study might be underpowered, and the value for Kappa could lie within a very wide range. Was the null hypothesis for the t-test that Kappa differs from a value of zero, or some other value you deemed to be a clinically relevant value for the agreement? Did you choose a one or 2-tailed test? It is important to comment upon this, or maybe quoting a confidence interval for Kappa would be better. You should consider removing the Kappa analysis from the study and simply report the values, if the statistical analysis adds no significant value. There is a helpful discussion about this at https://academic.oup.com/ptj/article/85/3/257/2805022
Discussion
Line 202: Clarify the explanation that “10.2% to 30.5% had asymmetries above 15%”. This is the range of asymmetries across all four categories of torque considered.
Line 220: I am not sure Arfaoui et. al. [36] is a good reference to show that differences in the application of forces don’t result in skin temperature asymmetries. That study did not measure the application of force in their cyclists, even though they found skin temperature across both legs to be similar. Is there a better reference you could cite?
Line 232: It is not very clear what is meant by the tskin being “unable to identify high percentages of negative-false and positive-false”. Are you saying that tskin has low positive and negative predictive values for predicting injury risk?
Line 239: Expand this discussion of the limitations of your study. By saying “the monitoring of muscle damage markers”, do you mean these could have been performed? Explain why you could not do this, and which would be interesting. Why is the age of the subjects a limitation? Would you recommend further research looking at temperature change during exercise? Presumably this would be an important follow up to your temperature studies at rest, because during exercise blood flow is recruited to the muscles, and temperature differences might become more evident.
Section 6 should simply be entitled “Author contributions”: I can see no detail in this section about patents.
Reviewer 3 Report
The manuscript addresses a relevant topic: the relation between strength muscular imbalances and skin temperature bilateral asymmetries. The authors applied the non-invasive technique Infrared Thermography (IRT), within a sample of 59 healthy soccer athletes. The manuscript consists of an interesting reading and well-structured.
The reviewer suggests few improvements, namely:
- In introduction, the authors could explore better the research gaps, before the definition of research objectives;
- The reviewer would like to suggest the reading of a recent paper about IRT, and the authors could cite in their review: Ana Colim, Pedro Arezes, Paulo Flores, Ricardo Vardasca & Ana Cristina Braga (2020) Thermographic differences due to dynamic work tasks on individuals with different obesity levels: a preliminary study, Computer Methods in Biomechanics and Biomedical Engineering: Imaging & Visualization, 8:3, 323-333, DOI: 10.1080/21681163.2019.1697757
- In tables, the authors should include a legend for the abbreviations (e.g. PL, NPL and AI…);
- In the discussion, the authors should explore better the limitations (related with the IRT technique), future work and practical applications of their research.
- Please, revise the format of the reference 28.
Round 2
Reviewer 2 Report
Congratulations on a much-improved manuscript.
I have two minor comments prior to publication:
1) In Table 4, "sensibility" should read "sensitivity"
2) Please consider working with the editorial office to shorten the "author constributions" section. This could be edited to improve claraity, and shortened without detriment to the paper.